# Benthic ecosystem cascade effects in Antarctica using Bayesian network inference

Emily G. Mitchell [1✉], Rowan J. Whittle[2] & Huw J. Griffiths [2]

Antarctic sea-floor communities are unique, and more closely resemble those of the Palaeozoic than equivalent contemporary habitats. However, comparatively little is known about the processes that structure these communities or how they might respond to anthropogenic change. In order to investigate likely consequences of a decline or removal of key taxa on community dynamics we use Bayesian network inference to reconstruct ecological networks and infer changes of taxon removal. Here we show that sponges have the greatest influence on the dynamics of the Antarctic benthos. When we removed sponges from the network, the abundances of all major taxa reduced by a mean of 42%, significantly more than changes of substrate. To our knowledge, this study is the first to demonstrate the cascade effects of removing key ecosystem structuring organisms from statistical analyses of Antarctica data and demonstrates the importance of considering the community dynamics when planning ecosystem management.

[1] Department of Zoology, University of Cambridge, Downing St, Cambridge CB2 3EJ, UK. [2] British Antarctic Survey, High Cross, Madingley Road, Cambridge CB3 0ET, UK. ✉email: ek338@cam.ac.uk

The ecological structure of modern Antarctic benthic marine communities differs from the rest of the world. There is a paucity of shell crushing predators (sharks, rays, durophagous decapods), leading to dominance by epifaunal suspension feeding groups such as sponges[1–5]. Disturbance to dominant benthic taxa, such as the sponges, has the potential to alter the ecosystem, but it is not known how. Generally, there are limited data on the structure and function of Southern Hemisphere benthic ecosystems compared with other areas globally. Despite this limited data, Antarctic benthic marine communities are abundant and diverse[6] with complex functional diversity and a highly structured, three-dimensional, physical configuration[1,7–9].

The South Orkney Islands (SOI) are located in the Southern Ocean, 604 kilometres north-east of the tip of the Antarctic Peninsula, at a latitude of around 60° 35′ S. The seafloor around the SOI has an exceptionally high biodiversity; one-fifth of the animals found in the entire Southern Ocean are represented[10,11]. Benthic biodiversity in the SOI has been studied previously[10–12]. Brasier et al.[10] showed that benthic communities in the SOI were strongly correlated with the hardness of the seafloor. Soft sediments were dominated by deposit feeders, hard surfaces had a greater abundance and richness of taxa, and an overall higher biomass. They were also dominated by filter feeding Vulnerable Marine Ecosystem (VME) taxa[10]. VME's are marine areas that are vulnerable to the effects of anthropogenic activities, such as fishing[13].

The South Orkney Islands Southern Shelf (SOISS) was the first designated Marine Protected Area (MPA) from the high seas on the planet (Fig. 1), established in 2009[10,14]. MPA's are clearly defined geographically, and managed using legal and protective measures for the long-term conservation of nature, associated ecosystem services, and cultural values[15]. The SOISS MPA is managed under the Commission for the Conservation of Antarctic Marine Living Resources (CCAMLR)[10]. Our data come

from within the South Orkney MPA and the surrounding area, at the shelf break. Using in situ photographs, ten dominant phyla can be identified from the SOISS benthic invertebrate community. The community is dominated by VME taxa (these are habitat forming taxa which could not be confidently identified from photographs), echinoderms and cnidarians, with frequent bryozoan and poriferan taxa[10].

New approaches for assessing biodiversity across MPA's are increasingly important given their increased prevalence in international conservation strategies and the associated costs of assessment and monitoring[10]. Our study determines the community structure on different spatial scales using Bayesian Network Inference (BNI) analysis. BNI is a technique to statistically infer the causal relationships (or dependencies) between different variables, which in this study are different taxa and different environmental factors such as substrate type and depth[16]. In the BNI presented here, the variables such as a taxon, or an environmental factor, are described as 'nodes', and the relationship between them as 'connections' (more usually known as 'edges' in network literature), so that the network consists of a series of connected nodes. The strength of the relationship between two nodes (i.e., the strength of the connection) is given by the Influence Score (IS), which is calculated as a cumulative frequency distribution over all possible discrete states[17]. Discrete Bayesian Network Inference Algorithms (BNIAs) enable the inference of causal relationships (over just mutual correlations), as well as non-linear interactions. We use BNIAs to determine the key taxa and environmental factors that underpin Antarctic benthic community dynamics and use the Bayesian networks to infer likely changes of taxon removal within this MPA.

The BNIA Banjo[16] was used to generate two different Bayesian networks over different spatial scales[18]. The fine-scale analysis used 527 individual photographs, which covered 0.51 m² for each sample and the large-scale analyses grouped photographs within

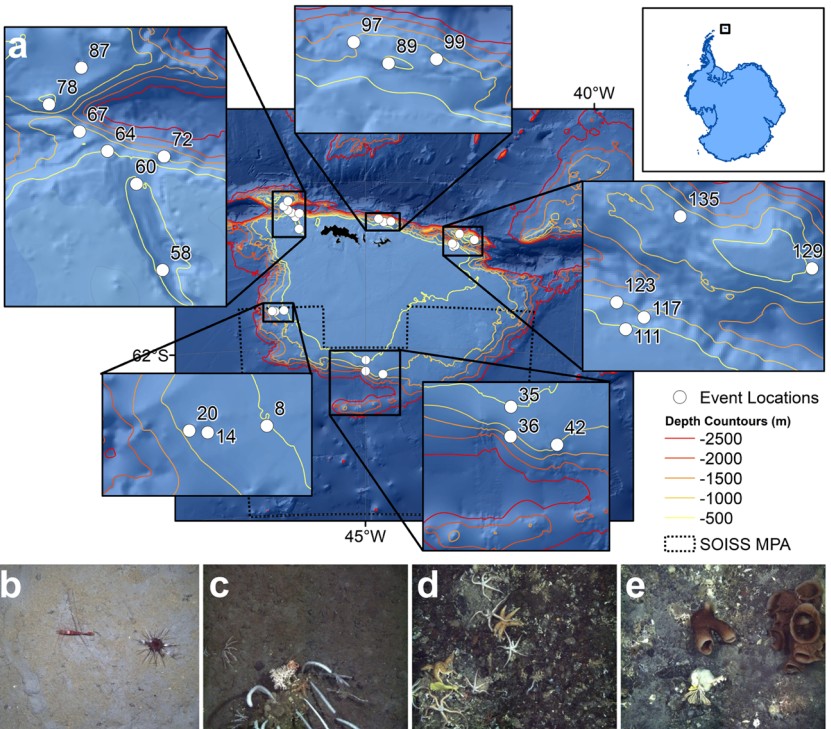

**Fig. 1 Locality map. a** Location map of South Orkney Islands sampling and **b–e** photographs of key taxa and habitats from the South Orkney Region. **b** Soft sediments dominated by echinoderms and arthropoda; **c** soft sediment with dropstones colonised by cnidaria, bryozoans, sponges, and encrusting taxa; **d** hard substrate dominated by echinoderms and encrusting taxa; **e** large sponge-dominated habitat.

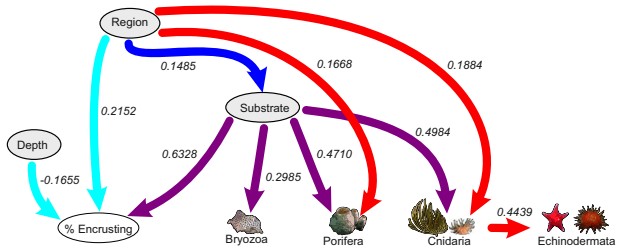

**Fig. 2 Fine-scale Bayesian network.** Dependencies between nodes are indicated by the lines connecting them, the width of which indicates the occurrence rate in the bootstrap analyses (wider lines indicate higher occurrence). Arrows indicate nonmutual dependence between two nodes where the head of the arrow is dependent taxa, for example the arrow from region to substrate indicates that substrate depends on region. Mutual dependencies are indicated as double headed arrows. Mean interaction strengths of the correlations are indicated; positive interaction strengths indicating aggregation, negative interaction strengths indicating segregation. Taxa that have no dependencies on another node are not shown. Purple arrows depict dependencies on substrate. The dark blue arrow is between two physical nodes (Region and Substrate). Red arrows are those dependencies resulting from Region and light blue arrows are those that affect percent encrusting.

21 transects of 1.2 km long for each sample. The raw data were filtered to avoid Type I errors through zero-inflation. The fine-scale analysis included the physical nodes: Depth, Substrate and Region (Fig. 1, Supplementary Table 1) and seven taxonomic groups (Supplementary Table 2). The large-scale analysis included Depth, Substrate and Region (Supplementary Table 3) and 12 taxonomic groups (Supplementary Table 4). The data were discretized into zero, low, and high groups[19], using medians to differentiate the boundary for the low and high abundances. For percentage encrusting and depth, quartiles were used, and modes for region and substrate (Supplementary Tables 5–6). Quartiles are used for percent encrusting and depth rather than medians because there is no zero state for these group, and so in line with previous work (e.g., Milns et al. 2010), we have used quartiles because just two states would limit the information that could be extracted from this factor, and three is not commonly used. To minimise outliers bias, 100 samples were bootstrapped at 95% level by randomly selecting 95% of the total number of samples for each analysis[20]. The subsample networks were then found using Banjo, and the final network taken to be the network where the connections occurred in the majority of bootstrapped sub-sampled with the strength of dependency given by the mean interaction strength of the bootstrapped connections (see 'Methods' for more details). This bootstrapping analysis was applied to both the fine-scale and large-scale data sets.

One of the most powerful aspects of using BNIA is the ability to make inferences of how one node (taxa or physical variable) is likely to change given another node being in a given state (zero, low, or high for a taxon). We used our BN to infer how the changes in abundance of the most connected taxon would affect the abundances of the other taxa.

## Results

**Fine-scale network**. Within the fine-scale network analyses (~0.51 m²), of the ten nodes included in the analyses, eight were connected within the network, with unidentified VME taxa and arthropods not connected. Substrate was the most connected node (five connections) and depth the least (one connection) (Fig. 2). There was one dependency between two taxa—Cnidarians and Echinoderms. The other dependencies were between physical nodes and taxa and two physical nodes.

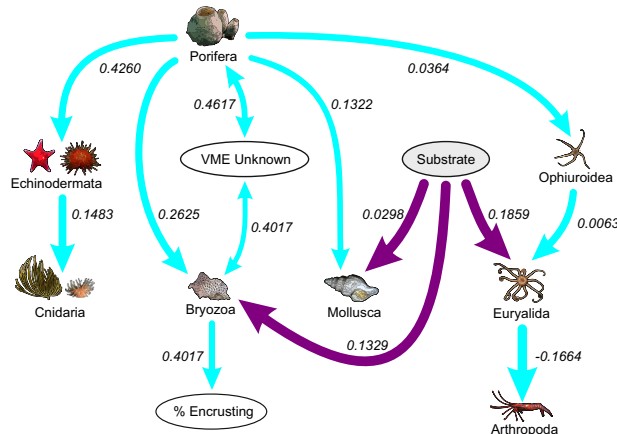

**Fig. 3 Bayesian networks of large-scale data.** Dependencies between nodes are indicated by the lines connecting them, the width of which indicates the occurrence rate in the bootstrap analyses (wider lines indicate higher occurrence). Single-headed arrows indicate nonmutual dependence between two taxa; mutual dependencies are indicated by double headed arrows. Mean interaction strengths of the correlations are indicated; positive interaction strengths indicating aggregation, negative interaction strengths indicating segregation. Taxa that have no dependencies with another node are not shown. Light blue depicts dependencies between different taxa and purple between physical nodes and taxa.

**Large-scale network**. The large-scale network (between sampling events) had 14 nodes within the analyses, of which Depth, Actinopteri (fish) and Annelids were not connected (Fig. 3). At this spatial-scale there were insufficient samples in the different Region groups to include Region in the analysis. Porifera was the most connected taxon (five connections) and substrate the most connected physical variable (three connections). The strongest dependency was between bryozoans and percentage of encrusting organisms (0.5639) and the only negative dependency was between euryalids (basket stars) and arthropods (−0.1664). All taxa had a direct or indirect connect with Porifera. There were two mutual dependencies between VME unknown and Porifera and between Bryozoa and VME unknown.

**Inferred taxon abundance changes with Porifera removal**. In the large-scale network Porifera is the most connected node and taxon (Fig. 3), and changes in their abundance will influence all taxa apart from the unconnected Annelida. The inferred changes in abundance state for all taxa connected to Porifera are given in Fig. 4 and Supplementary Table 7. For example when Porifera change from 'High' to 'Zero' abundance, the probability of Echinoderms existing in a 'Zero' abundance state increases by 80%, while the probability of 'High' state decreases by 60% (Fig. 4, Supplementary Table 7). This result suggests that if sponges decrease in abundance, the echinoderms will as well. This pattern of decreasing taxa abundance with decreasing Porifera is seen for all connected taxa except arthropods. Arthropods appear robust to changes, owing to negative dependency with Euryalida, which has a positive dependency with Porifera. With this being the smallest effect with a <20% change. Porifera have the greatest impact on Bryozoa, with a 100% change to zero state when Porifera are in zero state. Ophiuroidea see an increase in the probability of a 'Low' abundance state with 'Zero' Porifera state, but this is coupled to an increase in 'Zero' state probability, reflecting an overall abundance decrease. The strength of the effect of a decrease in Porifera on taxa decreases as the length of the chain of dependencies, number of nodes connecting the taxon to Porifera, decreases.

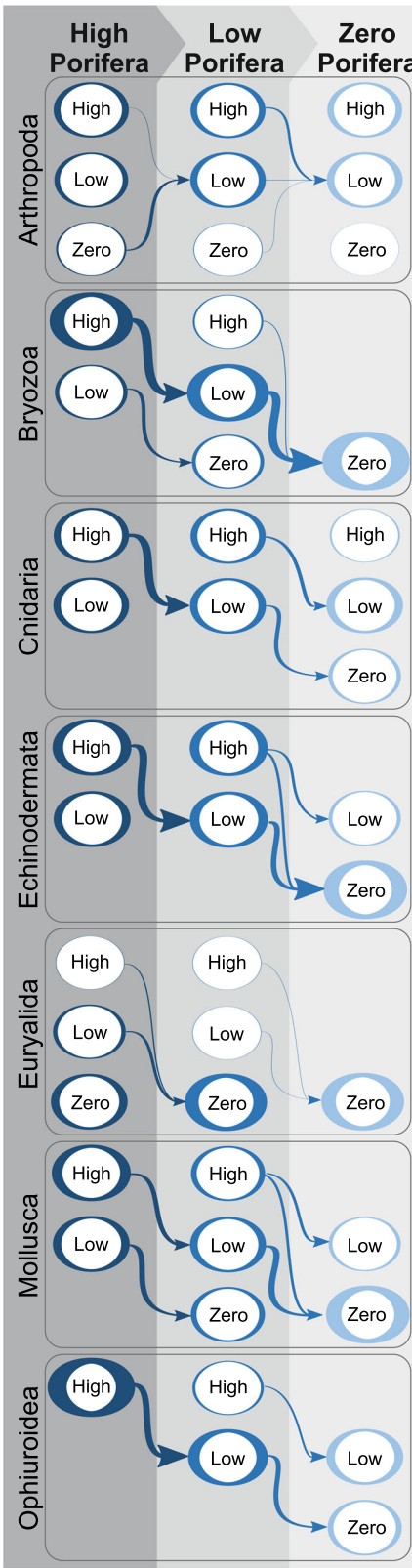

**Fig. 4 Schematic diagrams showing how the probabilities of each taxa being in a given state (zero, low, high) change given Porifera states.** The thickness of the arrow connecting different states indicates the percentage change between the two connected states. The thickness of the node indicates the proportion of samples in that state.

The effect of the changing abundance state of Porifera had a significant influence on the community, with a mean change in probability state (from High to Zero) $\Delta p^{Porifera} = 0.4210$. In contrast, the mean change in probabilities between different substrates is smaller (Supplementary Tables 7–9): between Silt to Rocky ($\Delta p^{Silt\ to\ Rocky} = 0.2181$), Silt to Gravel ($\Delta p^{Silt\ to\ Gravel} = 0.3569$) and Rocky to Gravel ($\Delta p^{Rocky\ to\ Gravel} = 0.1883$).

## Discussion

To our knowledge, this study is the first time the cascade effects of removing key ecosystem structuring organisms has been found with statistical analyses of data from a Marine Protected Area. By considering the community as a network of interactions, we were able to quantify the relative importance of different taxa. Porifera was the most connected variable within the large spatial-scale network and we were able to demonstrate the cascading effects that changing their abundance had on other taxa. These effects varied, depending on the proximity of each taxon to Porifera and the nature of the dependency, suggesting that different taxa have different resilience to the removal of sponges. However, the effects were mostly negative: a decrease in Porifera led to a decrease in echinoderms, molluscs, bryozoans and ophiuroids. Cnidarians had an intermediate dependency, and showed a smaller reduction. Notably, changing the abundance of Porifera had a stronger influence on the abundance of other taxa than changing the substrate type (Supplementary Tables 7–9), showing that the contribution of Porifera to the network is likely over and above them just being a colonisation substrate.

The only group that showed an increase in abundance with a decrease in sponges were the arthropods. However, only certain large and conspicuous arthropod taxa (decapods and large pycnogonids—Brasier et al.[10]) were detectable using the photographic analysis. The majority of images with high arthropod numbers were in areas of open, muddy habitat, leaving the individuals exposed and easier to count. Many smaller arthropods, such as isopods and amphipods (generally not detected using this scale of photography), are known to have strong positive ecological associations with Antarctic sponges, where the highest mean density of amphipods on a single sponge was 1295 individuals[21]. The predicted increase in arthropod numbers probably only applies to the observed large species that have a preference for open habitats, e.g., *Nematocarcinus lanceopes* and *Notocrangon antarcticus*.

Antarctic sponge-dominated communities are structured by biological interactions, supporting diverse microbial and macrofaunal communities[21,22]. Predation upon sponges is important in structuring Antarctic benthic communities[23,24]. They act as settling surfaces, contributing considerable structural heterogeneity to numerous colonising epibionts including benthic diatoms and bacteria[24–29]. Larger invertebrates such as polychaete worms, bivalves, gastropods, amphipods and isopods live on, and in sponges[24,30]. Sponges also directly affect the physical characteristics, structural and oceanographic, of the benthic habitat. Large demosponges and hexactinellids change the small-scale topography of the seafloor, providing new microhabitats and biogenic hard substrates. They also have the ability to transform the local oceanography by physically obstructing currents and, as active filter feeders, pumping thousands of litres of seawater through their systems and removing or concentrating nutrients[31].

Porifera can dominate some Antarctic seafloor communities. While forming a significant component of the community in our data, they are not the most numerous taxon, however, they do

dominate the biomass in most regions[10]. This study highlights the importance of sponges to the South Orkney ecosystem, as the organisms that have the most connections within our network and their significant influence on taxon abundance probabilities (Figs. 3, 4). Our result provided quantitative support for their importance as community structuring organisms, as highlighted by Bell[29].

Potential threats to the South Orkney ecosystem, in particular the sponges, include climate change, longline fishing and ocean acidification. Ocean warming is thought to cause increased thermal stress on sponge assemblages. This stress could cause disease and mortality owing to a decrease in the efficacy of defence mechanisms, and via the development of pathogens[32]. There are currently no commercial bottom fisheries within the South Orkney area, however, demersal fishing is occurring in other areas with similar habitats and ecology. Longline fishing for Patagonian toothfish occurs around South Georgia and other sub-Antarctic islands[14,33]. Research fishing for toothfish has also recently expanded to include a wider area, reaching to the north of the South Orkney Islands[14], putting the whole area at risk. At present the SOISS MPA covers <50% of the shelf and slope of the region meaning that the majority of the archipelago could be vulnerable to future fishing.

Our Bayesian network inference shows that disruption to the benthic community will cause cascade effects on many areas of the ecosystem, potentially leading to a regime shift from a rich and diverse epibenthic community supported by sponges, to an infaunal dominated system. The removal of sponges could result in a replacement by other habitat forming groups. However, our network analyses found that large branching or erect cnidarians did not mediate sponge loss and, given their physical structure, would be under similar threat from benthic fishing as the sponges[34]. Organisms with calcified skeletons, bryozoans and some cnidarians, are at an even greater risk from ocean acidification than sponges[35]; making the simple replacement of sponges with other similar groups unlikely, resulting in a marked change to both the habitat and community structure with the loss of the major three-dimensional structures.

As with all benthic image based analyses, many groups will be underrepresented. This underrepresentation is particularly true for the dominant infauna in soft sediment areas[10] (e.g., polychaetes and molluscs), and for organisms that are too small to capture in the image. This network method for understanding ecological change included a variety of data types, such as different physical variables, as well as biotic abundances. In order to delve deeper into ecosystems dynamics, the inclusion of more physical variables, such as temperature or pH, would enable further inference of how these communities are likely to change in the near future.

There are notable differences between the processes that regulate the fine and large-scale networks in our results. The fine spatial-scale networks were dominated by environmental variables, whereas the larger scale networks were dominated by taxa interactions. In the fine spatial-scale networks, substrate was the most connected node, and therefore changes in the substrate would have the greatest effect on the community. This reflects the findings of other studies around Antarctica[10,36,37]. Hard surfaces are linked to an abundance of Bryozoa, Porifera, Cnidaria and Echinodermata, owing to the potential for colonisation by a diversity of benthic organisms. There is only one taxonomic dependency at this fine scale—between Cnidaria and Echinodermata. Environmental factors, such as currents and hard surfaces, particularly on a small and medium scale, are conducive to the growth and diversity of suspension feeding assemblages. For dense aggregations of suspension feeders to occur, abundant suspended material, and water movement are needed[1]; hard surfaces are also required for colonisation. Areas where there are large accumulations of structural organisms, such as sponges, tend to be in more hydrodynamically active

regions[1], with surfaces suitable for colonisation. These requirements may be reflected by the influence of Substrate in our large-scale results. Antarctic benthic communities show a large amount of patchiness in species composition at small or intermediate spatial scales[1,7]. Our fine-scale analyses likely reflect the local environmental patchiness, at a between photograph scale, that result in these changes. The differences between the fine and large-scale networks demonstrate the importance of taking into account different spatial scales—without the large-scale network the between taxon interactions would not have been revealed.

To our knowledge, this study provides the first statistical evidence that changes to sponge-dominated communities will have cascading effects on most other aspects of Antarctic benthic community structure. This method provides a methodology to identify these habitat structuring taxa, in order to prioritise them for further protection, and for them to be considered in management plans for potentially harmful anthropogenic activities such as longline fishing. The dominance of sponges in driving our networks show that some VME taxa are more vital in maintaining the overall health of an ecosystem than others. Given these results, sponges should be considered as a priority, even above other VME taxa, when designing new conservation measures in Antarctica and the Southern Ocean.

## Methods

The data for this study were collected in the austral summer of 2016, on board the BAS research ship RRS James Clark Ross[38]. Biological abundance data was taken from photographic images from Brasier et al.[10] (see methodology within). The images were taken with a Shallow Underwater Camera System (SUCS), in transects of 10 photographs, 10 m apart. Replicant transects were separated by 100 m, at water depths of 500 m, 750 m and 1000 m[10]. Each photograph in the analysis was 0.51 m² in area. In the analyses, 'VME unknown' was biological material unidentifiable to phylum, but distinguishable as VME taxa, e.g., branched or budding fragments which could be bryozoan or cnidarian species. The percentage cover of encrusting species was recorded by Brasier et al.[10], as the number of individual colonies was not always possible to distinguish between colonising taxa[10]. Physical variables, such as substrate texture, were also observed from the SUCS images[10].

Taxonomic resolution of the data originally collected was dependent upon the ability to distinguish the organism in the images[10]. For our analysis, the taxonomic hierarchy analysed related to the abundance level of that group, as lower abundance, finer scale taxonomic groupings would be zero-inflated so not able to be used within our methodology (cf. Milns et al.[39]). Taxonomic groupings included were Annelida, Arthropoda, Cnidaria, Echinodermata, Mollusca, Bryozoa, Actinopteri, Porifera and Echinodermata. The echinoderm taxa Ophiuroides and Euryalids also occurred in large enough numbers to be analysed as separate groups. The taxa in each of these groupings can be seen in the Brasier et al.[10].

The raw data (taxon ID from photographs) was highly zero-inflated (84.9% of entries), so data grouping was performed to capture the fine-scale (individual photographs at ~cm scale) and large-scale (replicate transects at ~km scale) ecology of the ecosystems. For the fine-scale network there were 527 samples (abundances from individual photographs) and twelve nodes. Four were physical nodes (Depth, Region, Substrate and Substrate Texture, Table 1), two were functions of the specimens present (percent encrusting), five were taxa classified to Phylum level (Arthropoda, Bryozoa, Porifera, Cnidaria and Echinodermata) and there was one bin-group (VME Unknown) Supplementary Table 2.

For the large-scale network, the data from all the photographs in each event (up to three replicate transects) was combined to form 21 samples with 16 nodes. For factor nodes (Region, Texture and Substrate) the modal variable was taken, taxa nodes were summed, and the mean was used for Percent Encrusting and Depth. Texture was excluded from analyses due to the high zero-count. Two nodes were functions of the specimens present (percent encrusting), ten taxa were analysed (Annelid, Arthropod, Cnidarian, Echinoderm, Mollusc, Bryozoan, Actinopteri, Ophiuroidea, Euryalida, Porifera) and one bin-group (VME Unknown) was used. The combination of the two networks enabled the investigation of both physical and biotic interactions across multiple spatial scales.

We chose these relatively coarse taxonomic groupings in order to maximise the statistical power of our analyses. This statistical power enabled us to reconstruct a complex network of dependencies, thus revealing the subtle relationships between different taxa. The coarse grouping is a limitation of this method, because the homogenisation of some diverse groups, such as echinoderms, groups together organisms with variable life modes and traits (and therefore differing relationships with other taxa). However, the coarse level of identification was necessary with this volume of data (over 500 photographs), and the patchiness and rarity of many taxa. Future studies involving a higher density of photographs will allow for genus or species level separation, and/or an analysis based upon functional traits classification.

**Table 1 Table of network properties for the two networks found. Chains are defined as series of dependencies containing more than two nodes. Link density is mean number of connections per node, connectance is the number of connections/number of nodes*(number of nodes-1).**

| Network | mean IS | Nodes | Positive dependencies | Negative dependencies | Mutual | Connectance | Link density | Maximum chain length | Number of chains | Mean chain length |
|---|---|---|---|---|---|---|---|---|---|---|
| Fine-scale | 0.3225 | 8 | 9 | 1 | 0 | 0.179 | 1.25 | 3 | 5 | 2.2 |
| Large-Scale | 0.2272 | 11 | 14 | 1 | 2 | 0.154 | 1.55 | 5 | 10 | 2.7 |

**Analysis**. One approach to understanding how ecosystems function is to consider the ecosystem as a network (cf. Miln et al.[38]). Different taxa or groups of taxa are considered 'nodes' and their interactions are described as 'connections' (more usually known as 'edges' in Baysian networks), which link interacting taxa together. A node that the connections feed into are called 'parents'. Work has centred on gene regulatory networks[17], neural information flow networks, and with more recent applications to ecological and palaeoecological networks[18,39–41]. It is important to note that the structure produced by the BNIA reflects the associations caused by co-localisations (two taxa which both have a high abundance), not by a specific interaction, for example predation. By using BNIA, direct dependencies between taxa can be detected, minimising auto-correlation between two nodes. For example, if A depends on B which depends on C, there could be a correlation between A and C. However, this correlation would not represent an interaction or association between A and C, merely the two correlations between A and B and B and C. BNIA enables only the realised dependencies to be found, ensuring only actual interactions and associations are found.

Bayesian network inference was performed in Banjo[16], The BNIA software used was Banjo v2.0.0, a publicly available Java based algorithm[39,41]. Banjo uses uniform priors, so boundaries for the different discretized groups were chosen to ensure even splitting between the groups. Discretized data were input into Banjo, which then generated a random network based on the input variables. A greedy search was then performed to find a more likely network than the random one generated. This search was repeated 10 million times for each set of input data and the most probable network was then output. The maximum number of parents was set to 3 to limit artefacts[19].

The BNIA used require discrete data, which ensures data noise is masked, and only the relative densities of each taxon are important[39]. We split the data into three intervals; zero counts, low counts and high counts. Low counts consisted of counts below the median for the taxa group and high counts were counts over the median. Medians were used over means because for some groups the high counts were very high, and would result in a very small number of samples grouped in the highest interval (cf. Milns et al.[39]). A large amount of bins maintains the amount of information present in the dataset, while fewer bins provide more statistical power, and greater noise masking. Yu[19] has shown that for ecological data sets three different bins is a good balance. Zero was treated as a separate entity because the presence of one individual is very different to a zero presence, in contrast to zero gene expression, for example. Data preparation for Banjo (grouping and discretization) was carried out in R[42], as was the statistical analysis of the data. Further analysis of banjo outputs, when required, used the functional language Haskell[43]. The scripts are available on Github (github.com/egmitchell/bootstrap).

To minimise outliers bias, 100 samples were bootstrapped at 95% level by randomly selecting 95% of the total number of samples for each analysis[20,43], and then finding the subsample network using Banjo. For each connection calculated, the probability of occurrence was calculated, and the resultant distributions analysed to find the number of Gaussian sub-distributions using normal mixture models[44]. This probability distribution was bimodal for each data set, which suggests that there were two distributions of connections, those with low probability of occurrence, and those with highly probable connections. The final network for each area was taken to be those connections which were highly probable. The threshold for being labelled 'highly probable' depended on the network (as determined by the normal mixture modelling analyses): 53% for fine-scale network and 51% for the large-scale network. The magnitude of the occurrence rate is indicated in the network by the width of the line depicting the connection.

The direction of the connection between nodes indicates which node (taxon) has a dependency on the other node (taxon). For each connection, the directionality was taken to be the direction that occurred in the majority of bootstrapped networks. Where there was no majority (directional connections have a probability between 0.4 and 0.6), the connection was said to have bi-directionality, or indicated a mutual dependency.

The IS can be used to gauge the type and strength of the interaction between two nodes. If the IS = 1, this corresponds with a positive correlation. When node 1 is high, node 2 will be high. An IS of −1 corresponds to a negative correlation: High node 1 corresponds to a low node 2. An IS = 0 does not mean there is no correlation between the two nodes. IS = 0 means that the interaction is non-monotonic. Sometimes node 1 will be positively correlated with node 2, sometimes negatively. The mean IS for each connection was calculated for each sample area.

**Contingency test filtering**. In order to avoid Type I errors introduced by high zero counts, which is common in ecological data sets, we excluded rare taxa, which were found in under 33% of the grid cells. Note, that this method of exclusion could potentially mean that a taxon with high abundance in a very limited area is excluded from analyses. To further guard against Type I errors, we also used a method of contingency test filtering that removed from consideration a connection between two variables whose joint distribution showed no evidence of deviation from the distribution expected from their combined marginal distributions (chi-squared tests, $p > 0.25$)[39]. This threshold was used to ensure no chance of removing truly dependent dependencies, so that only artefacts such as those found between high zero counts were removed from consideration. These links were provided to the BNIA to exclude from consideration.

**Inference**. Inferring how one node (taxa or physical variable) is likely to change given another node being in a given state is done by calculating the probability of node A being in a given state given node B is in a given state. All nodes that have dependencies between A and B and so are included in the calculation:

$$P(A|B) = \sum_{n=1}^{n=N-1} \frac{\prod_{s=1}^{s=S} P(B_n | A_{n+1})}{\sum_{m=1}^{N} P(B | A_m) P(A_m)}$$

The $N$ are the total number of nodes in the chain and $n$ and $m$ are the indices for the chain of nodes of length N connecting the first and end nodes. S are the number of discrete states for each node, which are indexed $s$. In order to infer the likely change of one taxon's (A) abundance on another's abundance (B), the probabilities of all taxa that occur in the network between A and B have to be taken into account. For example, for the probability that B is in a Zero abundance state given A is in a high abundance state, and given that B is connected through a dependency of B on C and C on A, the probability is calculated as follows: the probabilities of C existing in all states is calculated given a High abundance state for A, and then the probabilities for B existing in a Zero state is calculated for each state of C and then summed together to give the probability of B in Zero given A in high. To work out the inferred change in B given a change in A from High to Zero would involve taking the difference between the probabilities for each state of B given A is high with each state of B given A is low. For the code used to generate these inferred probabilities, please see ref. [45].

**Reporting summary**. Further information on research design is available in the Nature Research Reporting Summary linked to this article.

## Data availability
Data generated within the study was modified from Braiser et al.[10] and is available in the public repository Figshare[46]: https://doi.org/10.6084/m9.figshare.12214568.v1.

## Code availability
Code is available on github.com/egmitchell/bootstrap and ref. [45]: https://doi.org/10.5281/zenodo.3969969.

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

## Acknowledgements

Natural Environment Research Council Independent Research Fellowship NE/S014756/1 to E.G.M. R.J.W. and H.J.G. are part of the British Antarctic Survey Polar Science for Planet Earth Programme; R.J.W. in the BAS Palaeoenvironments, Ice-sheets and Climate Change team and HJG in the Biodiversity, Evolution and Adaptation team. We thank Dr Madeline Brasier for analysing and quantifying the original photographs, the officers and crew of the RRS James Clark Ross and the participants in the SO-AntEco expedition (JR15005). This work is a contribution to the SCAR (Scientific Committee on Antarctic Research) AntEco (State of the Antarctic Ecosystem) Programme.

## Author contributions

E.G.M., R.J.W., and H.J.G. conceived the study, discussed the results, wrote the text, and made the figures. E.G.M. analysed the data. H.J.G. led the team collecting the original data.

## Competing interests

The authors declare no competing interests.
