## [Peer Review File · Communications Biology]

Reviewers' comments:

Reviewer #1 (Remarks to the Author):

The main objective of the study was to determine dependencies between the physical environment and main benthic taxa and across taxa, and to assess the importance especially of sponges structuring considerably the benthos. It is well known that sessile suspension feeders are abundant in the Antarctic benthos but this might be the first study to provide advanced statistical evidence about their relevance for other taxa to develop and grow, be reduced or get locally extinct. Despite the study is carried out "only" in a restricted area the results could -with great care- be applied to a much larger area, maybe even to a circumpolar scale. Therefore I consider the results of the manuscript to be important in both, a general biological context and for assessing the relevance of Marine Protected Area(s). The drawback of the manuscript is the bit confusing language using a mix of not well-defined terms -at least not in the main text- variables, nodes, edges. A more simple language would help understanding the general approach without reading the Method paragraph. The inconsistency of the study area/sites and the MPA (only 6 of 21 sites) must be much better explained or avoided by changing the justification of the study (slightly). A major part of the main objective are the consequences of sponge removal, which must be mentioned in the introduction. Since this study is based "only" on a modelling approach the general value of the results would even increase if a short critical review of the method would be added, e.g. how realistic are the results using this method and in which point or when and why could developments or conditions in nature deviate from the results of the models.

Line 41: Even if few aggregations are known, in general Antarctic macrobenthic communities are not (at all) dominated by stalked crinoids and brachiopods.

Line 43: Ophiuroids are known to be abundant also in benthic systems where crushing predators exist.

Line 46: Another paper describing epibiotic relationships, where sponges play an important role would be: Gutt J, Schickan T 1998. Epibiotic relationships in the Antarctic benthos. *Antarct Sci* 10: 398-405.

Maybe this paper is a more recent alternative to no. 7.: Gutt J, Arndt J, Kraan C, Dorschel B, Schröder M, Bracher A, Piepenburg D 2019. Benthic communities and their drivers: a spatial analysis off the Antarctic Peninsula. *Limnology and Oceanography* 64: 2341-2357; doi: 10.1002/lno.11187

Line 62: How is "The SOISS benthic community" defined? I assume that more than 10 phyla exist in the area.

Title, abstract, introduction, discussion: South Georgia should be mentioned, in the introduction and discussion also that the sampling sites are situated at the shelf break (only).

Fig. 1, title, abstract, introduction, discussion: 15 stations are not located in the MPA and only six stations at the northwestern margin of the MPA. This doesn't justify to focus in this study mainly on the MPA as indicated in the title, abstract and introduction.

Line 103: VME taxa?

Line 70: What are variables, network nodes or a dependency between two nodes or depth, substrate, region or also taxa?

Line 121: Which taxon/factor depends on which. The factor where the arrow points depends on that where the line starts (no arrow) or vice versa?

Discussion: To understand the approach of removing sponges (why only sponges?) two sentences explaining the approach is necessary already in the main text and not only in the second part of the manuscript in "Methods". This could also even be mentioned in the objectives of the study in the introduction, where the description of the objective(s) is quite short and too general.

Figs 4 and 5: Since there is redundancy between information provided in both Figs I would present in the main text only Fig 5 and shift Fig 4 to the Supplementary Material

Reviewer #2 (Remarks to the Author):

The authors considered the community in an Antarctic Marine Protected Area as a network of interactions and quantified the relative importance of different taxa using Bayesian analysis. It is a good thought and also an importance pathway to understand ecosystem function in future. I cannot evaluate the application of Bayesian analysis, because I do not know this method well. I am trying to give comments from an ecological view, hope editor and authors find it is useful to improve this manuscript. A few questions raised up when I read the manuscript, most are on methodology.

1) See Fig. 1 and Fig. 2, data are from underwater images taken along transects. They represents different habitats in the South Orkney Islands: soft sediment, hard substrate. The diverse habitats are key background to determine the diversity of communities. Thus, do you have a geographic habitat map in the studied area to show, e.g., proportions of soft sediment, sandy and hard substrate? Habitat priority, combining the sampling sites in your study, jointly determine the accuracy in Fig. 2.

2) See Fig. 3. It is the most important figure after Fig. 2 to show Bayesian networks of large-scale data. In this study, the taxonomic hierarchy analysis is related to the abundance level of that group. The abundance is individuals of images, right? If that so, the authors mentioned limitation in the methodology (Line 336-339). This is also the reason I have not enough confidence to Fig. 2.

Thus, I thought Bayesian network analysis as a method is deserved to further develop, but the current results cannot give me enough confidence to trust it is an ecological conclusion in the South Orkney Islands.

Reviewer #3 (Remarks to the Author):

Manuscript No.: COMMSBIO-20-1235-T

Title: Ecosystem cascade effects inferred using Bayesian networks in an Antarctic Marine Protected Area

1. General:

This manuscript built up fine-scale and large-scale networks using the Bayesian Network Inference Algorithms (BNIA) based on 527 photographic samples (for fine-scale) and 24 samples (for large-scale) in Antarctic marine. Further, the change of probability for different abundance status of each taxon with the removal of sponges was depicted. This manuscript is well-written and well-organized. I applaud the general approach and central question of the paper. However, there are a number of issues that need to be addressed before I can enthusiastically support its publication, especially the major points listed below.

2. Majors:

-My major concern is about the overfitting problem of the Bayesian network of large-scale data, considering there are only 24 samples but 16 variables. I understand the Bayesian networks could be robust against overfitting, but the paper could benefit by showing how the large-scale network is robust, considering the following sponges-removal analyses are all based on this large-scale network.

-Besides, I think the physical taxa (Region, Depth, and Substrate) should always be set as parent nodes (i.e. the arrows should always point from but not to the physical taxa in the network). For example, in Fig. 3 there are arrows pointing to Region from VME Unknown and Annelida. How to interpret it? It is very hard to image that the Region (N, NE, NW, S, and W) could be influenced by

the abundance of Annelida in the real world. A more logical explanation is that the abundance of Annelida could be influenced by Region. It is the same thing between Substrate and Bryozoa.

-There might be some error in Fig. 4, as the sum of the conditional probability for a taxon given a certain Porifera state should be 1. For example, the sum of the three probability values of Echnioderm being in zero state, low state, and high state under Low Sponge should be 1, considering there are only three states. However, in Fig. 4 it is $50\% + 50\% + 50\% = 150\% > 100\%$. Actually only a few taxa under certain conditions can meet this equality in Fig. 4 (e.g. Echnioderm under Zero Sponge, which is $80\% + 20\% + 0\% = 100\%$). Is this problem caused by the bootstrapping process? Besides, there is inconsistency between Fig. 4 and Fig. 5. I think these two figures generally describe the same thing, therefore one of them could go to Supporting Information (I would move Fig. 4 into SI). However, the inconsistency between the two figures confused me. For example, in Fig. 4, with High or Low Sponge, the probability of Echnioderm is generally even across the three states: Zero, Low, and High state. However, in Fig. 5, there are only two states and the Zero state disappear.

3. Minors:

-Line 78: m^2 should be $m^{>2}$.

-Line 84: I do not understand why you use "means" for Percentage Encrusting. I can totally understand why using "medians" for other taxa, but cannot we just also use "medians" for Percentage Encrusting? What is more, it is said in the table legends of Supplementary Tables 5 and 6 that Percent Encrusting is done using quartiles. I am totally confused. In Supplementary Table 4, there are four groups except for Zero. Are they separated by the quartiles? If so, why do you use quartiles for this taxon and divide it into five groups?

-Lines 85-90: Is this bootstrapping only applied in fine-scale or also in large-scale?

-Fig. 1: There are 21 sites in the map, but there are 24 samples for the large-scale networks. I wonder where the mismatch comes from.

-Lines 112-113: It states that Region is a categorical variable so IS cannot be interpreted literally. However, Substrate is also a categorical variable. How could you interpret that?

-Line 113: Define "IS" since it is the first appearance here. BTW, lines 398-404 describe IS, but it does not show how to calculate it. Is it calculated based on Pearson correlation? If so, how to deal with categorical variables such as Region and Substrate?

-Table 1: Some text in the head of the table does not show well (Mutual dependencies, and Mean Chain Length). Some terms are not defined, e.g. Connectance².

- Lines 175-176: The statement "The small-scale network showed no dependencies between taxa" is incorrect, because Echinodermate does show dependency on Cnidaria (Fig. 2). Besides, it is very hard to image there is rare relationship between biological taxa, since competition, predation or other interactions could lead to strong correlations. Could you discuss this result and give some potential reason?

that the biological taxa have little relationship

-Line 332: The right parenthesis is lack.

-Line 348: "The edges that feed into a node are called parents". Actually, not the edges but the nodes are parents.

-Line 431: there is an extra comma after the colon.

Reviewers' comments:

Reviewer #1 (Remarks to the Author):

The main objective of the study was to determine dependencies between the physical environment and main benthic taxa and across taxa, and to assess the importance especially of sponges structuring considerably the benthos. It is well known that sessile suspension feeders are abundant in the Antarctic benthos but this might be the first study to provide advanced statistical evidence about their relevance for other taxa to develop and grow, be reduced or get locally extinct. Despite the study is carried out "only" in a restricted area the results could -with great care- be applied to a much larger area, maybe even to a circumpolar scale. Therefore I consider the results of the manuscript to be important in both, a general biological context and for assessing the relevance of Marine Protected Area(s).

We thank the reviewer for recognizing the potential wider applications of the method and the use of this limited dataset as a test case.

The drawback of the manuscript is the bit confusing language using a mix of not well-defined terms -at least not in the main text- variables, nodes, edges. A more simple language would help understanding the general approach without reading the Method paragraph.

We have clarified the language at the start of the main text. Lines 72 - 80.

We have also clarified the language and made it more consistent, reducing the technical vocabulary to make it more accessible to a non-specialist audience e.g. "variables" have all been changed to "nodes" and "edges" has been replaced with "connections".

In BNI presented here the variables such as a taxon or an environmental factor are described as "nodes" and the relationship between them as "connections" (more usually known as "edges" in Bayesian networks) so that the network consists of a series of connected nodes. The strength of the relationship between two nodes (i.e. the strength of the connection) is given by the Influence Score (IS) which is calculated as a cumulative frequency distribution over all possible discrete states¹⁷. Discrete Bayesian Network Inference Algorithms (BNIAs) enable the inference of causal relationships (over just mutual correlations) as well as non-linear interactions.

The inconsistency of the study area/sites and the MPA (only 6 of 21 sites) must be much better explained or avoided by changing the justification of the study (slightly). A major part of the main objective are the consequences of sponge removal, which must be mentioned in the introduction.

The start of the introduction has been changed to address the focus of the study Lines 40-44:

'The ecological structure of modern Antarctic benthic marine communities differs from the rest of the world. There is a paucity of shell crushing predators (sharks, rays, durophagous decapods), leading to dominance by epifaunal suspension feeding groups such as sponges¹⁻⁴⁵. Disturbance to dominant benthic taxa, such as the sponges, has the potential to alter the ecosystem, but it is not known how.'

The title, introduction and abstract have been altered to clarify that the study is based on the South Orkney's and that some samples are within an MPA:

Title has been changed to *'Investigating benthic ecosystem cascade effects in Antarctica using Bayesian network inference.'*

South Orkney's added to abstract Lines 34 – 36.

'Our data comes from within the South Orkney MPA and the surrounding area, at the shelf break'
added to introduction

Since this study is based "only" on a modelling approach the general value of the results would even increase if a short critical review of the method would be added, e.g. how realistic are the results using this method and in which point or when and why could developments or conditions in nature deviate from the results of the models.

There has been some misunderstanding with the reviewer. This is primarily a statistical study, with the probabilities calculated using a statistical model, rather than a theoretical study. As such, our results are limited primarily by the input data rather than the methods. Several other studies have fully assessed the limitations of this methodology notably Smith et al. 2006, Yu et al. 2002, 2004 and Yu 2005, which is referenced in the methods section Lines 382 - 416 . The limitations of this study due to the data resolution are discussed on lines 281 - 288.

Line 41: Even if few aggregations are known, in general Antarctic macrobenthic communities are not (at all) dominated by stalked crinoids and brachiopods.

Changed on Lines 40 - 44 to highlights the importance of sponges and remove the confusion.

The ecological structure of modern Antarctic benthic marine communities differs from the rest of the world. There is a paucity of shell crushing predators (sharks, rays, durophagous decapods), leading to dominance by epifaunal suspension feeding groups such as sponges¹⁻⁵.

Line 43: Ophiuroids are known to be abundant also in benthic systems where crushing predators exist.

Ophiuroids deleted

Line 46: Another paper describing epibiotic relationships, where sponges play an important role would be: Gutt J, Schickan T 1998. Epibiotic relationships in the Antarctic benthos. *Antarct Sci* 10: 398-405.

Reference added

Maybe this paper is a more recent alternative to no. 7.: Gutt J, Arndt J, Kraan C, Dorschel B, Schröder M, Bracher A, Piepenburg D 2019. Benthic communities and their drivers: a spatial analysis off the Antarctic Peninsula. *Limnology and Oceanography* 64: 2341-2357; doi: 10.1002/lno.11187

Reference added

Line 62: How is "The SOISS benthic community" defined? I assume that more than 10 phyla exist in the area.

Changed for clarity on Lines 65 – 66.

Using in situ photographs, ten dominant phyla can be identified from the SOISS benthic invertebrate community.

Title, abstract, introduction, discussion: South Georgia should be mentioned, in the introduction and discussion also that the sampling sites are situated at the shelf break (only).

Title now reads '*Investigating benthic ecosystem cascade effects in Antarctica using Bayesian network inference.*'

South Orkney's added to abstract line 35 and the below to the introduction Lines 64 – 65.

Our data comes from within the South Orkney MPA and the surrounding area, at the shelf break.

Fig. 1, title, abstract, introduction, discussion: 15 stations are not located in the MPA and only six stations at the northwestern margin of the MPA. This doesn't justify to focus in this study mainly on the MPA as indicated in the title, abstract and introduction.

Title Changed (MPA removed)

Abstract reworded lines 34 – 36.

The South Orkney Islands, Antarctica, is an important ecosystem, as part of the locality is a Marine Protected Area.

Introduction lines changed lines 65 – 65.

Our data comes from within the South Orkney MPA and the surrounding area, at the shelf break'

We have left in the information about the SOISS MPA as it is relevant to the study. The ecosystem in the MPA and surrounding area are very similar, what affects one area will affect the other, therefore our results are of importance to the whole area, including the entire MPA.

Line 103: VME taxa?

'taxa' added on Line 119.

Line 70: What are variables, network nodes or a dependency between two nodes or depth, substrate, region or also taxa?

The variables are the nodes of the network so include both taxa and environmental factors such as depth. This has been clarified on Lines 72 – 77 and throughout when we are specifically referring to the nodes of the network we have changed variables to nodes.

BNI is a technique to statistically infer the causal relationships (or dependencies) between different variables, which in this study are different taxa and different environmental factors such as substrate type and depth¹⁶. In BNI presented here the variables such as a taxon or an environmental factor are described as "nodes" and the relationship between them as "connections" (more usually known as "edges" in network literature) so that the network consists of a series of connected nodes.

Line 121: Which taxon/factor depends on which. The factor where the arrow points depends on that where the line starts (no arrow) or vice versa?

We have clarified that on Lines 139-141.

Arrows indicate non-mutual dependence between two nodes where the head of the arrow is dependent taxa, for example the arrow from Substrate to Region indicates that Substrate depends on Region.

Discussion: To understand the approach of removing sponges (why only sponges?) two sentences explaining the approach is necessary already in the main text and not only in the second part of the manuscript in "Methods". This could also even be mentioned in the objectives of the study in the introduction, where the description of the objective(s) is quite short and too general.

We focused on the importance of removing sponges because our analysis found that they were the most connected taxa, which suggests that they are the most important. If we had found other taxa to be more connected we would have calculated this alternative. We have added a clarification of this point in the main text Lines 104-107.

One of the most powerful aspects of using BNIA is the ability to make inferences of how one node (taxa or physical variable) is likely to change given another node being in a given state (zero, low or high for a taxon). We used our BN to infer how the most changes in abundance of the most connected taxon would affect the abundances of the other taxa.

In methods Lines 442 – 443.

Inferring how one node (taxa or physical variable) is likely to change given another node being in a given state is

Figs 4 and 5: Since there is redundancy between information provided in both Figs I would present in the main text only Fig 5 and shift Fig 4 to the Supplementary Material

Agreed, we have moved Fig 4 to the SI and also changed it to table format (now Table S4), since in its current form it caused some confusion.

Reviewer #2 (Remarks to the Author):

The authors considered the community in an Antarctic Marine Protected Area as a network of interactions and quantified the relative importance of different taxa using Bayesian analysis. It is a good thought and also an importance pathway to understand ecosystem function in future. I cannot evaluate the application of Bayesian analysis, because I do not know this method well. I am trying to give comments from an ecological view, hope editor and authors find it is useful to improve this manuscript. A few questions raised up when I read the manuscript, most are on methodology.

1) See Fig. 1 and Fig. 2, data are from underwater images taken along transects. They represents different habitats in the South Orkney Islands: soft sediment, hard substrate. The diverse habitats are key background to determine the diversity of communities. Thus, do you have a geographic habitat map in the studied area to show, e.g., proportions of soft sediment, sandy and hard substrate? Habitat priority, combining the sampling sites in your study, jointly determine the accuracy in Fig. 2.

The substrate composition by transect is available in Supplementary Information Tables 1 & 3 and also covered in detail by Brasier et al, 2018 (Fig 2).

2) See Fig. 3. It is the most important figure after Fig. 2 to show Bayesian networks of large-scale data. In this study, the taxonomic hierarchy analysis is related to the abundance level of that group. The abundance is individuals of images, right? If that so, the authors mentioned limitation in the methodology (Line 336-339). This is also the reason I have not enough confidence to Fig. 2.

Our network analysis is a way of quantifying the co-occurrence of all of the inputs (physical or biological). It does not consider taxonomic hierarchy and treats all variables independently and equally (including physical parameters such as substrate or depth). The absolute abundances do not feature in the analyses, instead it is the relative abundances within each sample that are the focus on this work.

Thus, I thought Bayesian network analysis as a method is deserved to further develop, but the current results cannot give me enough confidence to trust it is an ecological conclusion in the South Orkney Islands.

This is not a method that we have developed, there are multiple publications using these techniques (e.g. Milns et al. 2010, Mitchel and Butterfield 2018), however we appreciate the concerns of the reviewer and the limitations of the study presented and list many of them in the discussion. We believe that given the novel application of this technique to a little understood system it was sensible to start with a coarse taxonomic resolution for this study, especially as computational costs would increase dramatically with higher resolution data. We believe that the results are both interesting and meaningful and are comparable with previously published ecological studies using more established techniques e.g, Brasier et al 2018. Future work will build upon this demonstration of the method.

Reviewer #3 (Remarks to the Author):

Manuscript No.: COMMSBIO-20-1235-T

Title: Ecosystem cascade effects inferred using Bayesian networks in an Antarctic Marine Protected Area

1. General:

This manuscript built up fine-scale and large-scale networks using the Bayesian Network Inference Algorithms (BNIA) based on 527 photographic samples (for fine-scale) and 24 samples (for large-scale) in Antarctic marine. Further, the change of probability for different abundance status of each taxon with the removal of sponges was depicted. This manuscript is well-written and well-

organized. I applaud the general approach and central question of the paper. However, there are a number of issues that need to be addressed before I can enthusiastically support its publication, especially the major points listed below.

2. Majors:

-My major concern is about the overfitting problem of the Bayesian network of large-scale data, considering there are only 24 samples but 16 variables. I understand the Bayesian networks could be robust against overfitting, but the paper could benefit by showing how the large-scale network is robust, considering the following sponges-removal analyses are all based on this large-scale network.

There is extensive study trying to quantify how best to balance sample size, number of discrete groups and number of parents to ensure robust networks while minimizing Type 1 and Type 2 errors (see Smith et al. 2006, Yu et al. 2002, 2004 and Yu 2005). Using these previous studies, we balance the number of discrete groups, variable and sample numbers with the Banjo settings changing parameters such as the maximum number of parents depending to minimize errors. None-the-less if/when there is severe over fitting, then the IS approaches 1. For example, in the original data we had two nodes, Substrate and Texture where texture was whether the substrate was hard or soft. Banjo found a mutual dependency between these variables with a $IS=0.9967$. We therefore excluded this variable from analysis since it did not provide new insights. The mean IS for the two networks (Table 1) suggests that overfitting is not generally a problem.

Network robustness can be detected by looking at a) the mean value of IS and b) the distribution of edges found with the bootstrapping. When the network is not robust, then most IS scores approach zero and/or there are not many edges which occur a high proportion of the time with bootstrapping. For the fine-scale network the distribution of notes is given below, there are a great many edges that are found under 20% of the time, and relatively few that are found >50% of the time. It is these high-frequency edges that are taken to be the network nodes cf. Milns et al. 2010.

-Besides, I think the physical taxa (Region, Depth, and Substrate) should always be set as parent nodes (i.e. the arrows should always point from but not to the physical taxa in the network). For example, in Fig. 3 there are arrows pointing to Region from VME Unknown and Annelida. How to interpret it? It is very hard to image that the Region (N, NE, NW, S, and W) could be influenced by the abundance of Annelida in the real world. A more logical explanation is that the abundance of Annelida could be influenced by Region. It is the same thing between Substrate and Bryozoa.

We agree that the Region edges are problematic, and thank the reviewer for pointing out this issue. We re-ran the analyses with the edges fixed as suggested for substrate and Region (network given below). The network remains the same as the previous one, with slightly different IS due to the fact that they are calculated as means of a bootstrap, so are expected to vary.

Banjo Version 2.0.1\nHigh scoring network, score: -204.6452\nProject: sucsEvents7\nUser: emily mitchell\nDataset: sucsEvents7.txt\nNetworks searched: 13000000

However, the power of this technique is in reconstructing ecological relationships without any external input (such as by setting node directionality), so that we think it is preferable to not set the nodes for these variables.

We note that the two connecting edges for Region were weak (0.007 and 0.0604) suggesting only a very weak association, so given our preference for not imposing any relationships on the data, we instead thought that makes more sense to remove the Region node from the analyses, especially since some of the sample numbers for Regions are low for some areas. With Region removed from analyses, Annelids are not included in the network, with the other edges remaining.

Banjo Version 2.0.1\nHigh scoring network, score: : -178.4053\nProject: sucsEvents7\nUser: emily mitchell\nDataset: sucsEvents7.txt\nNetworks searched: 13000000

We have updated the Figure Fig 3, and Table 1 to reflect this new network.

-There might be some error in Fig. 4, as the sum of the conditional probability for a taxon given a certain Porifera state should be 1. For example, the sum of the three probability values of Echnioderm being in zero state, low state, and high state under Low Sponge should be 1, considering there are only three states. However, in Fig. 4 it is $50\% + 50\% + 50\% = 150\% > 100\%$. Actually only a few taxa under certain conditions can meet this equality in Fig. 4 (e.g. Echnioderm under Zero Sponge, which is $80\% + 20\% + 0\% = 100\%$). Is this problem caused by the bootstrapping process? Besides, there is inconsistency between Fig. 4 and Fig. 5. I think these two figures generally describe the same thing, therefore one of them could go to Supporting Information (I would move Fig. 4 into SI). However, the inconsistency between the two figures confused me. For example, in Fig. 4, with High or Low Sponge, the probability of Echnioderm is generally even across the three states: Zero, Low, and High state. However, in Fig. 5, there are only two states and the Zero state disappear.

We apologize that this figure is confusing. The probabilities need to total one given the Taxon state rather than the Sponge state. We have moved this information to the Supplementary information and think it would be clearer as a table. The columns and rows of former Figs 4 and 5 now match clearly with Table S7.

3. Minors:

-Line 78: m2 should be m2.

-Line 84: I do not understand why you use “means” for Percentage Encrusting. I can totally understand why using “medians” for other taxa, but cannot we just also use “medians” for Percentage Encrusting? What is more, it is said in the table legends of Supplementary Tables 5 and 6 that Percent Encrusting is done using quartiles. I am totally confused. In Supplementary Table 4, there are four groups except for Zero. Are they separated by the quartiles? If so, why do you use quartiles for this taxon and divide it into five groups?

Our apologies for this confusion due in part to mistakes in the manuscript. Quartiles are used for % encrusting rather than medians because there is no zero state for this group. In keeping in with previous work we have used quartiles because just two states would limit the information that could be extracted from this factor, and three is not commonly used. We have corrected this on Lines 93-94. The zero state wasn't assigned so we have corrected this in Table S4.

We have clarified these points on Lines 94-97.

Quartiles are used for percent encrusting and depth rather than medians because there is no zero state for these group, and so in line previous work (e.g. Milns et al. 2010) we have used quartiles because just two states would limit the information that could be extracted from this factor, and three is not commonly used.

-Lines 85-90: Is this bootstrapping only applied in fine-scale or also in large-scale?

Our apologies for this oversight we bootstrapped both the fine and large scale, and have corrected on Lines 103.

This bootstrapping analysis was applied to both the fine-scale and large-scale datasets.

-Fig. 1: There are 21 sites in the map, but there are 24 samples for the large-scale networks. I wonder where the mismatch comes from.

-Lines 112-113: It states that Region is a categorical variable so IS cannot be interpreted literally. However, Substrate is also a categorical variable. How could you interpret that?

We did not discuss a literal translation of Substrate, because while it is a categorical variable it is on a range of soft to hard. However, because these categories were qualitatively assigned, it is

most consistent to treat it as a categorical variable, and so have added in this comment on Lines 130-131.

Geographic 'region' and Substrate are categorical variables, with no correlation between numerical factor and place for Region, so the signal of the IS cannot be interpreted literally. The Substrate category changes from soft to hard, so the IS is roughly interpreted as a trend with decreasing hard substrate.

-Line 113: Define "IS" since it is the first appearance here. BTW, lines 398-404 describe IS, but it does not show how to calculate it. Is it calculated based on Pearson correlation? If so, how to deal with categorical variables such as Region and Substrate?

Our apologies for this mistake. We have defined the influence score IS on Lines 79-80. It is defined in Yu et al. 2004 as a cumulative frequency distribution over possible discrete states and its calculation is fully described in Yu et al. 2004 and Milns et al. 2010.

The strength of the relationship between two nodes (i.e. the strength of the edge) is given by the Influence Score (IS) which is calculated as a cumulative frequency distribution over all possible discrete states.

-Table 1: Some text in the head of the table does not show well (Mutual dependencies, and Mean Chain Length). Some terms are not defined, e.g. Connectance2.

Our apologies, these have been corrected on Lines 173 and we have defined Connectance and Connectance two on lines 174 - 176.

Link Density is mean number of edges per node, Connectance is the number of connections/NumberOfNodes² and Connectance2 is the connections/NumberOfNodes(NumberOfNodes-1)*

- Lines 175-176: The statement "The small-scale network showed no dependencies between taxa" is incorrect, because Echinodermate does show dependency on Cnidaria (Fig. 2).

Our apologies for this mistake, we have corrected this on Lines 189-192.

The small-scale network only had a single dependency between taxa, between Cnidaria and Echinoderms (Fig. 2), so that only Echinoderm abundance has the possibility of changing if the abundance Cnidaria changed.

Besides, it is very hard to image there is rare relationship between biological taxa, since competition, predation or other interactions could lead to strong correlations. Could you discuss this result and give some potential reason?

that the biological taxa have little relationship

The spatial scale is key here - we are looking at very small spatial scales (< 1 m), so it isn't so much about whether or not there are biological relationships between the taxa, but whether they are at these spatial scales. We can see that the taxa have a number of relationships between each other over the large spatial scales. For example, if the environmental variations are larger spatial scales then any inter-taxa interactions due to them will not be captured in this fine-scale analysis.

-Line 332: The right parenthesis is lack.

Corrected line 315.

-Line 348: "The edges that feed into a node are called parents". Actually, not the edges but the nodes are parents.

Our apologies, of course this is correct, and we have corrected our mistake on Lines 371-372.

The node that the edges that feed into are called parents

-Line 431: there is an extra comma after the colon.

Removed

Reviewers' comments:

Reviewer #1 (Remarks to the Author):

In the revision of the manuscript the authors considered the very most of my critical points and solved the problems. I also checked the responses to the comments of the two other reviewers. I am not able to assess all methodological detail criticized by referee #3 but I do not see major reasons anymore not to publish this manuscript.

Julian Gutt

Reviewer #3 (Remarks to the Author):

See attached document.

Manuscript No.: COMMSBIO-20-1235A

Title: Investigating benthic ecosystem cascade effects in Antarctica using Bayesian network inference

The revised manuscript has been improved a lot. I appreciate the authors' efforts and my major concerns have been addressed. However, there are still some minor comments.

-Figure 1: The authors did not reply to my question about the mismatch between the 21 sites in the map and the 24 samples in the large-scale networks.

-Lines 140-141: Is the term "the arrow from Substrate to Region" correct? From Figure 2 it is actually from Region to Substrate.

-Table 1: I am afraid some of the values are incorrect. I listed them below:

- Nodes: It should be 11 for the large-scale network.
- Connectance: Based on the definition, the Connectance for fine-scale network should be $(9+1)/8^2 = 10/64 = 0.156$, and for large-scale network should be $(14+1)/11^2 = 0.124$. I think it is meaningless to show this Connectance because it is used for the network with self-loops (i.e. an edge that connects a node to itself). Clearly there is no self-loop here, so the authors can just show Connectance2 (and rename it as Connectance).
- Connectance2: The definition should be the number of connections / [NumberOfNodes * (NumberOfNodes - 1)]. Based on the definition, the Connectance2 for the fine-scale network should be $(9+1)/(8*7) = 0.179$, and for the large-scale network should be $(14+1)/(11* 10) = 0.136$.
- Link Density: For the large-scale network, LD should be $(14+1)/11 = 1.364$.
- Maximum Chain Length: Isn't it 4 for the large-scale network?
- Number of Chains and Mean Chain Length: Please check the values, which I think might be incorrect, especially for the fine-scale network.

We thank the Reviewer 3 for their attention to detail, and finding our mistakes which have now been rectified below.

-Figure 1: The authors did not reply to my question about the mismatch between the 21 sites in the map and the 24 samples in the large-scale networks.

Our apologies, this is a typo, now corrected to 21 lines 87, 349.

-Lines 140-141: Is the term “the arrow from Substrate to Region” correct? From Figure 2 it is actually from Region to Substrate.

Our apologies, corrected lines 140-141

-Table 1: I am afraid some of the values are incorrect.

Thank you for catching these, and our apologies for not finding them ourselves. We corrected/addressed them below.

I listed them below:

– Nodes: It should be 11 for the large-scale network.

– Connectance: Based on the definition, the Connectance for fine-scale network should be $(9+1)/82 = 10/82 = 0.122$, and for large-scale network should be $(14+1)/112 = 0.124$. I think it is meaningless to show this Connectance because it is used for the network with self-loops (i.e. an edge that connects a node to itself). Clearly there is no self-loop here, so the authors can just show Connectance2 (and rename it as Connectance).

– Connectance2: The definition should be the number of connections / [NumberOfNodes * (NumberOfNodes - 1)]. Based on the definition, the Connectance2 for the fine-scale network should be $(9+1)/(8*7) = 0.179$, and for the large-scale network should be $(14+1)/(11*10) = 0.136$.

*Thank you for noticing these errors and the suggestion, which we have made. However, the mutual connections (of which there are two for the large-scale network) mean that the connectance is $(14+1+2)/(11*10) = 0.154$*

– Link Density: For the large-scale network, LD should be $(14+1)/11 = 1.364$.

This needs to be corrected for the correct number of nodes, but is $(14 + 1 + 2)/11 = 1.55$ (includes the mutual dependencies).

– Maximum Chain Length: Isn't it 4 for the large-scale network?

We have included loops when calculating Chain Lengths, so the 5 connection chain is:

Substrate -> Bryozoa -> VME Unknown -> Porfiera -> Bryozoa -> % Encrusting

– Number of Chains and Mean Chain Length: Please check the values, which I think might be incorrect, especially for the fine-scale network.

Corrected with our apologies.